# The Malocclusion Impact Questionnaire (MIQ): Cross-Sectional Validation in a Group of Young People Seeking Orthodontic Treatment in New Zealand

**DOI:** 10.3390/dj7010024

**Published:** 2019-03-04

**Authors:** Philip E. Benson, Fiona Gilchrist, Mauro Farella

**Affiliations:** 1Academic Unit of Oral Health, Dentistry and Society, School of Clinical Dentistry, University of Sheffield, Sheffield S10 2TA, UK; f.gilchrist@sheffield.ac.uk; 2Department of Oral Sciences, Faculty of Dentistry, University of Otago, Dunedin 9054, New Zealand; mauro.farella@otago.ac.nz

**Keywords:** orthodontics, malocclusion, quality of life

## Abstract

The aim of the study was to test the validity of the Malocclusion Impact Questionnaire (MIQ) in a NZ sample and to evaluate possible cross-cultural differences in MIQ data between a NZ and a UK sample. A cross-sectional, non-random sample of young people, aged 10–16 years, attending their first appointment at the orthodontic clinic of New Zealand’s National Centre for Dentistry were asked to complete a questionnaire. This consisted of the 17 item MIQ, the short form CPQ_11-14_-ISF16 and two global questions. Some basic demographic and clinical data were collected. Sixty-six participants completed the questionnaire; however, the data for 2 were excluded due to the number of incomplete responses. MIQ was found to have excellent internal consistency (Cronbach’s alpha 0.924), good construct validity (Spearman’s rho, 0.661 global Q1 ‘Overall, how much do your teeth bother you?’; 0.583 global Q2 ‘Overall, how much do your teeth affect your life?’). MIQ also demonstrated good criterion validity with CPQ_11-14_-ISF16 (Pearson rho, 0.625). The Rasch analysis confirmed that the questionnaire performed similarly and there was no differential item functioning between the two populations. The main differences between the samples were that the young people in NZ were less concerned about their malocclusion and reported lower item-impact scores compared with the young people in the UK.

## 1. Introduction

Locker and Allen [1] defined oral health-related quality of life (OHQoL) as “the impact of oral disorders on aspects of everyday life that are important to patients and persons, with those impacts being of sufficient magnitude, whether in terms of severity, frequency or duration, to affect an individual’s perception of their life overall”. Several studies have shown that malocclusion can impact on a child’s oral health quality of life (OHQoL) [2,3,4].

A number of generic measures of OHQoL have been developed for use in children [5,6,7] and several studies using these measures have shown that malocclusion can impact on a child’s OHQoL [2,3,4]. However the meaning and significance of these measures have recently been questioned [1], particularly for conditions such as malocclusion.

Generic measures are useful for comparing OHQoL between different conditions, whereas specific measures focus on problems relevant to that condition or disease, making them more sensitive [8], more acceptable to participants, and therefore, higher completion rates are more readily achievable. Their specific nature makes them more likely to respond to change [9].

A condition-specific measure has recently been developed to assess the impact of malocclusion on young people [10,11]. A cross-sectional evaluation of the new measure was undertaken in a sample of 184 young people attending a new patient appointment in a UK dental teaching hospital. This study concluded that the criterion and construct validity, internal reliability/consistency and test–retest reliability of the MIQ were good but that further testing in different populations was required to assess the generalizability of the measure.

The aim of this study was to undertake a further evaluation of the questionnaire in a population of young people attending an orthodontic clinic in New Zealand. The specific objectives were to test the construct and criterion validity, the internal consistency in a New Zealand population, and to compare the performance of the questionnaire with the original UK sample, using item-impact and Rasch analysis.

## 2. Methods

Ethical approval was obtained from the Human Ethics Committee (Health), University of Otago (ref: H16/009; 1 Feb 2016). The methods used in NZ were the same as the UK study. The design was cross-sectional, using a non-random, convenience sample. Participants were recruited when they attended for a new patient appointment at a dental teaching hospital, the Orthodontic Clinic, Faculty of Dentistry, University of Otago, Dunedin. There are no eligibility criteria for referral to the clinic, but the study inclusion and exclusion criteria were as follows:

The inclusion criteria were young people:aged 10–16 years;either gender and any ethnic group;who described themselves as “needing a brace”.

The exclusion criteria were young people with a:history of previous orthodontic treatment;severe skeletal discrepancy or a cleft of the lip and/or palate;complex medical history or learning disability that would impair understanding of the measure.

Potential participants and their parents were invited to take part in the study after their initial consultation, the purpose of which was described in general terms. The young people and their parents were given separate written information sheets, as well as the questionnaire. The young person was encouraged to complete the questionnaire on their own and return it at their initial visit, for example, whilst waiting to have diagnostic radiographs. If this was not possible then they were asked to take the questionnaire away, complete it at their convenience and return it in a pre-paid envelope, which was provided.

Each questionnaire consisted of a front sheet, which was detached and completed by the clinician, containing the participant’s allocated study number and summary details of their occlusion. The participant was given the rest of the measure with their participant study number, to self-complete, starting with their demographic details (age and gender), followed by two global questions ‘Overall, how much do your teeth bother you?’ and ‘Overall, how much do your teeth affect your life?’ This was followed by the 17-item Malocclusion Impact Questionnaire (MIQ) and the 16-item short form of the Child Perceptions Questionnaire (CPQ_11-14_-ISF16). Finally, there was a box to allow participants to enter any free text comments, asking if there was anything else they would like to add from any of the questions they had answered.

### Data Analysis

The response format for the two global questions was a 5-point severity scale and was scored from 0 = ‘Not at all’ to 4 = ‘Very much’. MIQ consists of a 3-point severity scale with scores 0 = ‘don’t’; 1 = ‘a bit’ and 2 = ‘very’ or ‘a lot’ for negatively worded questions (‘Nervous’, ‘Shy’) and 0 = ‘very’ or ‘a lot’; 1 ‘a bit’ and 2 = ‘don’t’ for positively worded questions (‘Happy’, ‘Good looking’, ‘Confident’). The wording of the response format was based on the Child Health Utility 9D index (CHU9D), which is a generic measure of HRQoL, developed specifically for children [12]. The scores for each item were added together to obtain a total score, the minimum score being 0 and maximum score being 34; higher scores indicated poorer OHQoL.

CPQ_11-14_-ISF16 has been used previously to assess the OHQoL in young people aged 11–14 years [13]. The questionnaire consists of 16 items organized into four subscales (oral symptoms, functional limitations, emotional well-being and social well-being). The response format is a 5-point frequency scale and ranges from 0 = ‘Never’ to 4 = ‘Everyday/almost everyday’. The scores for each item are added together to obtain a total score. The minimum score is 0 and the maximum score is 64, and again, higher scores indicate a greater frequency of impacts, and hence poorer OHQoL.

The data from the New Zealand participants were entered into an Excel spreadsheet (v 2016, Microsoft Corp, Washington, US). If a participant did not respond to more than 8 items, the entire questionnaire was excluded from the analysis. When 8 or fewer responses were missing, each absent value was substituted with the mean for the individual [14]. The NZ data were compared with previously reported data from a sample of UK participants [11].

The five global responses were collapsed into three categories (‘Not at all’ and ‘A little’; ‘Somewhat’; ‘Quite a bit’ and ‘Very much’). The frequencies and proportions of participants responding in the three categories were compared with that of the UK sample. The descriptive statistics for the total MIQ, as well as the total CPQ_11-14_-ISF16 and four domains were calculated. The correlation between the total MIQ and the total CPQ_11-14_-ISF16, and the two global questions was assessed using a Spearman’s rank correlation to determine construct validity. The correlation between the total scores of MIQ and the total scores of CPQ_11-14_-ISF16 was assessed using a Pearson product correlation coefficient to determine the criterion validity. Cronbach’s alpha was used to test the internal consistency/reliability. The item-impact scores for each MIQ question were calculated by multiplying the proportion of participants indicating a moderate or significant impact (scores of 1 or 2) with the mean sample score for that question. The item-impact scores for each CPQ_11-14_-ISF16 question were calculated by multiplying the proportion of participants indicating a moderate or significant frequency of impact (scores of 2, ‘Somewhat’; 3, ‘Often’ and 4, ‘Everyday or almost everyday’) with the mean sample score for that question. Statistical tests were undertaken using SPSS (v24 IBM Corp, NY, USA).

The fit and function of the MIQ questions were examined using an item response theory Rasch model [15]. In addition, the items were assessed to ensure they were free from differential item functioning (DIF). That is, that they function in the same way between the two populations. The methods used was the unrestricted or partial credit model suggested by Tennant and Conaghan [16]. The Rasch analysis was undertaken using RUMM2030 (RUMM Laboratory Pty Ltd, WA, Australia).

## 3. Results

### 3.1. Demographics and Occlusion

A summary of the demographic and occlusal data for the NZ participants compared with the previously described demographic and occlusal data for the UK sample is shown in Table 1.

The ratio of males and females were very similar between the two groups and confirms previous findings that a higher proportion of females tend to seek orthodontic treatment. The NZ sample had a slightly higher proportion of participants in the younger age groups and there was a higher proportion of participants with a class II division 1 incisor relationship and a lower proportion with class II division 2 and class III incisor relationships compared to the UK sample. There was also a higher proportion of participants with moderate and severe upper arch crowding and severe crowding in the lower arch in the UK sample.

### 3.2. Descriptive Analysis

Two participants had more than 8 missing responses and were excluded from the rest of the analysis. One had 8 missing responses and five had one missing responses. For these participants the missing responses were replaced with the participant’s mean.

Table 2 shows the numbers and proportions of participants responding in the three collapsed categories for the two global questions.

Nearly three quarters of the NZ participants responded that their teeth had little or no effect on their life overall and just over two thirds suggested that their teeth bothered them little or not at all. This is compared to over half of the UK sample responding that their teeth bothered them ‘quite a bit’ or ‘very much’.

Table 3 has the descriptive data for the responses of participants in the two groups to the CPQ_11-14_-ISF16 and MIQ questionnaires.

Both the median and mean total CPQ_11-14_-ISF16 and MIQ scores were higher in the UK sample and these were statistically significant (Mann-Whitney U, p < 0.001). The mean CPQ_11-14_-ISF16 functional domain score was slightly higher in the UK group, but not statistically significant (Mann-Whitney U, p = 0.201). The main differences were that the UK participants reported higher median and mean scores in the emotional and social well-being domains (Mann-Whitney U, p < 0.001). There were no ceiling effects for either the total CPQ_11-14_-ISF16 or total MIQ scores, but there were floor effects for MIQ with six individuals in the NZ group recording a total score of 0. There was only one recorded score of 0 in the UK group.

### 3.3. Validity and Reliability

The correlation between the participant scores for the global question, ‘Overall, how much do your teeth bother you?’ and the total MIQ score was 0.661 (Spearman’s rho, p < 0.001), and the total CPQ_11-14_-ISF16 was 0.523 (Spearman’s rho, p < 0.001). The correlation between the scores for the global question, ‘Overall, how much do your teeth affect your life?’ and the total MIQ score was 0.583 (Spearman’s rho, p < 0.001), and the total CPQ_11-14_-ISF16 was 0.434 (Spearman’s rho, p < 0.001). This suggests that MIQ has good validity for these two constructs.

The correlation between total CPQ_11-14_-ISF16 and the total MIQ scores was 0.625 (Pearson rho, p < 0.001) showing that MIQ has good criterion validity with a commonly used measure of OHQoL in young people.

Cronbach’s alpha was 0.924 for MIQ and 0.782 for CPQ_11-14_-ISF16, showing that MIQ has excellent internal consistency.

### 3.4. Item-Impact Analysis

The item-impact scores for each of the CPQ_11-14_-ISF16 questions are shown in Table 4.

The item-impact scores were considerably higher for the UK responses to the questions in the emotional and social well-being domains. The scores were also higher for the UK responses to the questions in the functional domain, but not for those in the oral symptoms domain.

Table 5 shows the item-impact scores for each of the MIQ questions.

This demonstrates that the item-impact scores were higher for the responses by UK participants to all the questions.

### 3.5. Rasch Analysis

All items demonstrated good fit to the model with fit residual in the range ±2.5, and summary chi square of P = 0.06. There was no DIF between the NZ and UK populations, suggesting that the questions function similarly in both populations. The mean person location for the NZ participants is −2.64 compared to −1.22 for those in the UK when the items are centred on zero (Figure 1). The upper section of the graph shows the distribution of participants and the lower part shows the distribution of thresholds (category transitions) of the items. The x-axes display the location (severity of impact) of the participants and the item location (difficulty) of the item thresholds. The y-axes show the frequency of item thresholds and participants. The graph confirms the finding that the NZ participants reported fewer impacts than their UK counterparts.

## 4. Discussion

The currently available measures of OHQoL were designed to assess the everyday impacts of oral disorders on people’s lives. Some questions in these generic measures are not relevant to people with malocclusion, and other specific problems relating to malocclusion are not included [17]. The Malocclusion Impact Questionnaire (MIQ) has therefore been developed, and initially validated in the UK, as a malocclusion-specific measure. The data from this study suggest that although the questionnaire performed similarly with young people in New Zealand as with young people in the UK, there were some differences between the two samples.

The main difference was that the young people in New Zealand were less concerned about their teeth than those in the UK. According to the responses to the global questions, the majority reported that their teeth had little or no effect on their life overall and that their teeth bothered them little or not at all. This difference might be because the NZ participants were slightly younger than the UK participants, with a higher proportion of 10 and 11 year olds.

One reason for the higher proportion of younger participants in the NZ sample might be due to differences in the methods of funding orthodontic treatment between the two countries. In the UK, orthodontic treatment for young people 16 years and under, like most dental treatment is free and the cost is covered by the National Health Service. In New Zealand there is no government finance for orthodontic treatment and parents have to pay. The dental school in Dunedin charges for orthodontic treatment, but the fee is lower than that in private specialist practice. The dental school is therefore a popular place to receive orthodontic treatment and the waiting list for treatment is long. It is possible that general dentists are aware of this and they, therefore, refer young people early, before they have lost all their primary teeth, because they are expecting a long wait for treatment.

The correlations between the global questions and MIQ were slightly reduced in the NZ sample compared with the UK sample, suggesting a moderately strong, rather than a strong association, and therefore, construct validity (‘Overall, how much do your teeth bother you?’ NZ Spearman’s rho = 0.661, UK Spearman’s rho = 0.733; ‘Overall, how much do your teeth affect your life?’ NZ Spearman’s rho = 0.583, UK Spearman’s rho = 0.701). Similarly, the correlation between the total CPQ_11-14_-ISF16 and the total MIQ scores was slightly lower (NZ Pearson rho = 0.625, UK rho = 0.751). In contrast the Cronbach’s alpha for the MIQ was slightly higher in the NZ sample (0.924) than in the UK sample (0.906).

The findings for CPQ were similar (‘Overall, how much do your teeth bother you?’ NZ rho = 0.523, UK rho = 0.720; ‘Overall, how much do your teeth affect your life?’ NZ Spearman’s rho = 0.434, UK Spearman’s rho = 0.589), however, the correlations and the Cronbach’s alpha (NZ α 0.782; UK α 0.841) were lower for CPQ_11-14_-ISF16 than for MIQ. This suggests that the MIQ performs better in terms of criterion validity and internal consistency than CPQ_11-14_-ISF16 when assessing the impact of malocclusion.

This concurs with the findings of Marshman and colleagues [17], who undertook a qualitative study to explore the face and content validity of CPQ_11-14_-ISF16. They found that several aspects of the measure, including the response format, the use of ‘double’ questions and the interpretation of certain words were confusing and not considered relevant to young people with malocclusion. The participants in this study also expressed the view that several issues caused by their malocclusion were not included in the measure; hence the need to develop a condition-specific measure.

One of the limitations of this study was that it was a convenience sample. All the orthodontic clinics at Dunedin, where new patients were expected, were accessed over a 7-week period. There is no reason to believe that the patients attending these clinics were unrepresentative of patients referred to the dental school; however, it is possible that patients referred to the one dental teaching hospital in New Zealand are not representative of those referred for orthodontic treatment in New Zealand, the great majority of whom attend private specialist practices. Nearly all the young people and their parents who were asked to take part in the study did so. Those who were unable to complete the questionnaire at the time, usually had to leave because of other childcare commitments, agreed to take the questionnaire away, complete it at home and return it by post or e-mail. It was not possible to assess the test-retest reliability of the questionnaire in the New Zealand sample due to the time constraints of the fellowship.

The responsiveness of the MIQ or the ability to assess change in impacts due to orthodontic treatment, is currently being assessed in a longitudinal study. The ultimate objective of developing the questionnaire is to evaluate the effectiveness of orthodontic interventions with a patient-reported outcome, rather than the currently used clinician-derived outcomes. In addition, the measure might be used to ensure that treatment is carried out in those with high normative need, who suffer the most impacts on daily life. De Oliveira et al [18] found that children’s perceived need for orthodontic treatment was explained more by the use of a normative measure (IOTN) in combination with a measure designed to assess OHQoL, than by using IOTN alone. Patients and parents, as well as care providers (and commissioners where applicable) would benefit from the development of a triage screening measure for orthodontic provision to effectively allocate funding. This will give the optimal balance between meeting the orthodontic needs of those who might most benefit and allocating resources to other more pressing dental health problems.

## 5. Conclusions

The MIQ was found to perform similarly in a sample of young people referred for orthodontic treatment to a New Zealand dental teaching hospital, compared with a UK dental teaching hospital; although the young people in the New Zealand sample were less concerned about the appearance of their teeth than the young people in the UK sample. Stronger associations between the global questions and the condition-specific measure total scores suggest that this measure has greater criterion validity in young people with malocclusion than the generic measure total scores.

## Figures and Tables

**Figure 1 dentistry-07-00024-f001:**
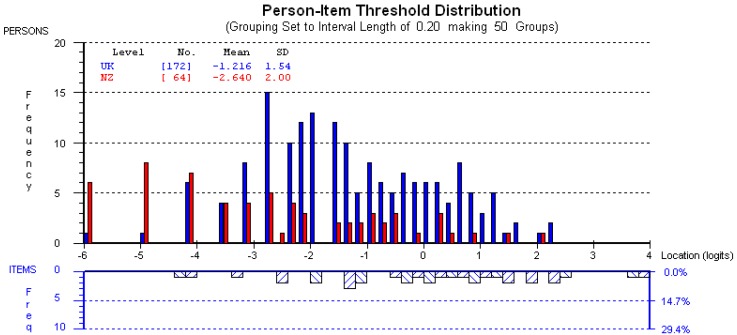
Graph showing the targeting of the MIQ in the NZ sample.

**Table 1 dentistry-07-00024-t001:** Demographics and occlusal data for NZ participants (n = 66) and UK (n = 184).

Demographic or Occlusal Characteristic		NZ	UK
Gender ^a^	Male	26 (40%)	71 (39%)
	Female	39 (60%)	113 (61%)
Age (yrs) ^b^	10	9 (14.1%)	11 (6.0%)
	11	15 (23.4%)	21 (11.4%)
	12	12 (18.8%)	40 (21.7%)
	13	12 (18.8%)	44 (23.9%)
	14	8 (12.5%)	43 (23.4%)
	15	5 (7.8%)	23 (12.5%)
	16	3 (4.7%)	2 (1.1%)
Incisor Relationship ^c^	Class I	24 (36.4%)	55 (30.1%)
	Class II division 1	29 (43.9%)	66 (36.1%)
	Class II division 2	5 (7.6%)	24 (13.1%)
	Class II intermediate	2 (3.0%)	7 (3.8%)
	Class III	6 (9.1%)	31 (16.9%)
Upper arch	Spaced	15 (22.7%)	43 (23.4%)
	No crowding or mild (0–4 mm)	41 (62.1%)	50 (27.2%)
	Moderate (5–8 mm)	8 (12.1%)	52 (28.3%)
	Severe (>8 mm)	2 (3.0%)	39 (21.2%)
Lower arch ^d^	Spaced	8 (12.1%)	19 (10.4%)
	No crowding or mild (0–4 mm)	43 (65.2%)	114 (62.6%)
	Moderate (5–8 mm)	14 (21.2%)	34 (18.7%)
	Severe (>8 mm)	1 (1.5%)	15 (8.2%)

^a^ data missing for 1 NZ participant; ^b^ data missing for 2 NZ participants; ^c^ one UK participant had missing lower incisors and no judgement was made about the incisor relationship; ^d^ data missing for 2 UK participants

**Table 2 dentistry-07-00024-t002:** Responses of the NZ participants (n = 64) and UK participants (n = 184) to the two global questions.

Global Question	‘Not at all’ or ‘A little’		‘Somewhat’		‘Quite a bit’ or ‘Very much’	
	NZ	UK	NZ	UK	NZ	UK
Overall, how much do your teeth bother you?	43 (67.2%)	60 (33.3%)	11 (17.2%)	23 (12.8%)	10 (15.6%)	97 (53.9%)
Overall, how much do your teeth effect your life?	47 (73.4%)	105 (58.7%)	12 (18.8%)	21 (11.7%)	5 (7.8%)	53 (29.6%)

**Table 3 dentistry-07-00024-t003:** Descriptive data for the Child Perceptions Questionnaire (CPQ_11-14_-ISF16) and the Malocclusion Impact Questionnaire (MIQ) responses for the NZ (n = 64) and UK (n = 184) participants.

Questionnaire and Domains		Median		Mean		SD		Min		Max	
		NZ	UK	NZ	UK	NZ	UK	NZ	UK	NZ	UK
CPQ_11-14_ ISF16	Oral symptoms	4	4	4.5	4.3	2.3	2.3	1	0	11	10
	Functional limitations	2	2	2.6	3.2	2.3	2.8	0	0	9	11
	Emotional well-being	1	4	2.0	5.0	2.5	4.3	0	0	11	16
	Social well-being	1	3	1.4	3.3	1.7	3.2	0	0	7	15
	Total score	8	14	10.5	15.8	6.4	9.5	2	1	27	47
MIQ	Total score	5	10	7.1	11.6	6.6	6.5	0	0	27	28

**Table 4 dentistry-07-00024-t004:** Item-impact scores for each CPQ_11-14_-ISF16 question (NZ, n = 64; UK, n = 184).

Domain	Question	NZ	UK
Oral symptoms	Pain in teeth	0.20	0.16
	Sores in mouth	0.07	0.18
	Bad breath	0.33	0.27
	Food stuck	1.02	0.53
Functional limitations	Longer eating	0.21	0.46
	Difficulty biting/chewing	0.15	0.25
	Difficulty words	0.01	0.04
	Difficulty drinking	0.19	0.29
Emotional well-being	Felt irritable/ frustrated	0.05	0.33
	Felt shy/embarrassed	0.07	0.59
	Been concerned other people think	0.19	0.80
	Been upset	0.02	0.31
Social well-being	Avoided smiling/laughing	0.06	0.50
	Argued with other children/family	0.06	0.22
	Teased	0.00	0.18
	Asked questions	0.02	0.16

**Table 5 dentistry-07-00024-t005:** Item-impact scores for each MIQ question (NZ, n = 64; UK, n = 184).

Question	NZ	UK
Happy	0.48	1.17
Good looking	0.59	1.20
Confident	0.42	0.89
Normal	0.18	0.65
Sad	0.06	0.26
Nervous	0.14	0.26
Shy	0.13	0.30
Smile	0.42	0.63
Laugh	0.07	0.39
Seeing photographs	0.26	0.51
Talk in public	0.05	0.18
Others nicer teeth	0.13	0.50
Being bullied	0.01	0.13
Making friends	0.01	0.04
Fitting in with friends	0.02	0.08
Cover with hand	0.03	0.18
Biting some foods	0.12	0.24

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
