# Peer review of "The Malocclusion Impact Questionnaire (MIQ): Cross-Sectional Validation in a Group of Young People Seeking Orthodontic Treatment in New Zealand"

_dentistry, 2019, doi:10.3390/dj7010024_

Round 1

Reviewer 1 Report

The article entitled The Malocclusion Impact Questionnaire (MIQ) - cross-sectional validation in a group of young people seeking orthodontic treatment in New Zealand is interesting however, it does not contribute significantly to the field.

Author Response

The article entitled The Malocclusion Impact Questionnaire (MIQ) - cross-sectional validation in a group of young people seeking orthodontic treatment in New Zealand is interesting however, it does not contribute significantly to the field.

Response: The reviewer has ticked the boxes indicating that the introduction, design, methods, results and conclusions must be improved, but does not give any details about how these are deficient.

Reviewer 2 Report

The Malocclusion Impact Questionnaire (MIQ) – cross-sectional validation in a group of young people seeking orthodontic treatment in New Zealand.

This is a well-written manuscript describing the validation of a new tool (MIQ) to quantitatively assess the impact of malocclusions on young people. It tests the questionnaire on 66 young people aged 10-16 years seeking orthodontic assessment at a university orthodontic clinic in New Zealand. The MIQ was compared to another existing tool, the Child Perceptions Questionnaire (CPQ11-14 ISF16) to assess validity, reliability and item impact. The results of the NZ group were compared with the results obtained from a group of 184 young people in the UK)

Overall, the findings indicate the MIQ has sound validity and consistency for constructs regarding how much teeth bother an individual and how much teeth affect an individual’s life. Also, the item impacts were consistent between the MIQ and CPQ11-14 ISF16, with the UK group reporting higher impacts of their malocclusion in all domains.

I have a few queries and suggestions that the authors could address before considering this manuscript ready for publication:

1.       Introduction: I think the authors could introduce the reader to the findings from the initial investigation of the MIQ with the UK sample and set the scene for why they want to further validate their tool in another setting. This could be kept brief but, without going back to the original paper, I was a bit lost as to what the aim of the study was and the key details of the original group.

2.       Methods: In order for an individual to be eligible for referral to the Orthodontic Clinic at the University of Otago, are there any key exclusion criteria for eligibility? Are only the most severe cases referred from the lowest socio-economic population groups? Perhaps a brief sentence would highlight if there are eligibility criteria.

3.       Methods: the last sentence ends with a question mark – perhaps this could be rewritten so the content of the question is paraphrased so it is not left as a question.

4.       Methods: I would like to see a brief description of the recruitment of the UK sample and if there were eligibility criteria for their inclusion in the study. How much does this sample differ from the NZ sample? Are they comparable? I understand this is discussed in the Discussion but some upfront description will help the reader better understand the background and methods for this study.

5.       Data analysis: Paragraph 2 – can any references be provided for previous use of the CPQ11-14 ISF16?

6.       Results: Demographics and occlusal data, and responses to the questionnaires – were the differences between the groups assessed statistically? Were they significantly different? This information could be reported in the tables.

7.       Figure 1: The contrast in colours for the graph is poor and I can’t tell which bar belongs to which group – could the design of this graph be improved and include a legend?

Author Response

1.      Introduction: I think the authors could introduce the reader to the findings from the initial investigation of the MIQ with the UK sample and set the scene for why they want to further validate their tool in another setting. This could be kept brief but, without going back to the original paper, I was a bit lost as to what the aim of the study was and the key details of the original group.

Response: Two sentences, providing more background about the findings of the initial investigation, have been added to the Introduction (Lines 47 to 51).

2.      Methods: In order for an individual to be eligible for referral to the Orthodontic Clinic at the University of Otago, are there any key exclusion criteria for eligibility? Are only the most severe cases referred from the lowest socio-economic population groups? Perhaps a brief sentence would highlight if there are eligibility criteria.

Response: A sentence has been added to the methods to make it clear that there were no eligibility criteria for referral to the clinic (Lines 61 & 62).

3.      Methods: the last sentence ends with a question mark – perhaps this could be rewritten so the content of the question is paraphrased so it is not left as a question.

Response: The question mark has been deleted, as the sentence was not really structured as a question.

4.      Methods: I would like to see a brief description of the recruitment of the UK sample and if there were eligibility criteria for their inclusion in the study. How much does this sample differ from the NZ sample? Are they comparable? I understand this is discussed in the Discussion but some upfront description will help the reader better understand the background and methods for this study.

Response: A sentence has been added explaining that the methods used in NZ were the same as the UK study (Line 59).

5.      Data analysis: Paragraph 2 – can any references be provided for previous use of the CPQ11-14 ISF16?

Response: The sentence has been amended and one study referenced that has used CPQ11-14 ISF16? Previously.

6.      Results: Demographics and occlusal data, and responses to the questionnaires – were the differences between the groups assessed statistically? Were they significantly different? This information could be reported in the tables.

Response: We thank the reviewer for this comment, but the primary objective of the study was to investigate the performance of the questionnaire between the two populations, not to look at differences in the responses between the two populations. We do compare the questionnaire scores between the two samples (lines 162 to 169). Concerning the participant demographics, the study was not powered to detect a difference between the two samples; therefore, interpretation of any null findings might be uncertain. We would prefer that readers look at the data in Table 1 and judge for themselves any similarities between the two samples.

7.      Figure 1: The contrast in colours for the graph is poor and I can’t tell which bar belongs to which group – could the design of this graph be improved and include a legend?

Response: We are sorry that the reviewer was unable to distinguish the colours in Fig 1. We wonder if they were looking at a low-resolution image, as the red and the blue are quite distinct in the original image. A figure legend has been provided.

Reviewer 3 Report

Interesting paper

Only suggestion is that in abstract they summarise the differences between NZ and Uk sample, instead of saying it is discussed.

Also , was there any measure of severity of malocclusion such as indices , IOTN,,

The uk sample had more cases with severe or moderate crowding and Class III & Cl II div 2 cases that explains the differences

Author Response

Interesting paper

Only suggestion is that in abstract they summarise the differences between NZ and Uk sample, instead of saying it is discussed.

Response: The abstract has been amended (Lines 28 to 30).

Also, was there any measure of severity of malocclusion such as indices , IOTN.

Response: We did not collect any information concerning orthodontic treatment need, but collected clinical data to determine that there was a range of malocclusions represented in the two samples. Previous studies have found a weak association between clinical symptoms, such as occlusal indices and OHQoL.

The uk sample had more cases with severe or moderate crowding and Class III & Cl II div 2 cases that explains the differences.

Response: We thank the reviewer for this comment, which we have described in the results (Lines 139 to 145).

Reviewer 4 Report

Nice study, very well presented, more suitable for an Orthodontics Journal.

Author Response

Thank you for this comment.

Round 2

Reviewer 1 Report

The authors could include pictures of Malocclusion to understand it better.

The study is based on a questionaire so it can debated because of the sample size.

The statistical analysis used should be more clearly described in the materials and methods section.

The authors should describe the hospitals that participated in the study.

Author Response

The authors could include pictures of Malocclusion to understand it better.

Response: We thank the reviewer for this comment, but do not believe that the addition of clinical images of malocclusion is required for this article.

The study is based on a questionaire so it can debated because of the sample size.

Response: Apologies, but I am afraid we do not understand this comment.

The statistical analysis used should be more clearly described in the materials and methods section.

Response: We would be grateful for more direction about how the description of the statistical analysis (highlighted in yellow) could be improved.

The authors should describe the hospitals that participated in the study.

Response: This is described at the beginning of the methods section and we have added some more additional detail, such as this was a dental teaching hospital in Dunedin (highlighted yellow).